# Two New Alkaloids and a New Butenolide Derivative from the Beibu Gulf Sponge-Derived Fungus *Penicillium* sp. SCSIO 41413

**DOI:** 10.3390/md21010027

**Published:** 2022-12-29

**Authors:** Yuxiu Ye, Jiaqi Liang, Jianglian She, Xiuping Lin, Junfeng Wang, Yonghong Liu, Dehua Yang, Yanhui Tan, Xiaowei Luo, Xuefeng Zhou

**Affiliations:** 1Institute of Marine Drugs, Guangxi University of Chinese Medicine, Nanning 530200, China; 2CAS Key Laboratory of Tropical Marine Bio-Resources and Ecology/Guangdong Key Laboratory of Marine Materia Medica, South China Sea Institute of Oceanology, Chinese Academy of Sciences, Guangzhou 510301, China; 3The National Center for Drug Screening, Shanghai Institute of Materia Medica, Chinese Academy of Sciences, Shanghai 201203, China; 4State Key Laboratory for Chemistry and Molecular Engineering of Medicinal Resources, School of Chemistry and Pharmaceutical Sciences, Guangxi Normal University, Guilin 541001, China

**Keywords:** sponge-derived fungus, *Penicillium* sp., PI3K, NF-κB

## Abstract

Marine sponge-derived fungi have been proven to be a prolific source of bioactive natural products. Two new alkaloids, polonimides E (**1**) and D (**2**), and a new butenolide derivative, eutypoid F (**11**), were isolated from the Beibu Gulf sponge-derived fungus, *Penicillium* sp. SCSIO 41413, together with thirteen known compounds (**3**–**10**, **12**–**16**). Their structures were determined by detailed NMR, MS spectroscopic analyses, and electronic circular dichroism (ECD) analyses. Butenolide derivatives **11** and **12** exhibited inhibitory effect against the enzyme PI3K with IC_50_ values of 1.7 μM and 9.8 μM, respectively. The molecular docking was also performed to understand the inhibitory activity, while **11** and **12** showed obvious protein/ligand-binding effects to the PI3K protein. Moreover, **4** and **15** displayed obvious inhibitory activity against LPS-induced NF-κB activation in RAW264.7 cells at 10 µM.

## 1. Introduction

The Beibu Gulf, a semi-closed gulf located in the northwest of the South China Sea, is rich in fishery or marine biological resources, including seagrass, coral, and sponge [1]. It has become an important source region of marine natural compounds [2]. The marine sponge, one of the most primitive and inferior multicellular animals, possesses abundant microorganisms germinated in its body and surface on account of its unique filter feeding system. Sponge-derived microorganisms act as an important guarantee for sponge survival since the lack of morphological physical defense structures [3,4,5].

In recent years, sponge-derived microorganisms have become one of the most abundant sources of new natural products, including terpenoids, alkaloids, sterols, peptides fatty acids, amino acids, and so on. Plenty of new secondary metabolites of sponge-derived fungi have been discovered with striking bioactive properties such as anti-tumor, antibacterial, antiviral, anti-inflammatory, and other biological activities [6]. Diketopiperazines are common secondary metabolites from a wide range of fungi, while quinazoline-containing diketopiperazines are relatively rare, with the difficulty in the determination of their configurations [7]. Butenolide derivatives, possessing the α,β-unsaturated γ-butyrolactone skeleton, were frequently isolated from fungi with diverse biological activities, especially the antitumor activity with diverse mechanisms or targets [8].

In our research for novel bioactive natural products from marine sponge-derived fungi, the strain fungus *Penicillium* sp. SCSIO 41413 was isolated from a *Callyspongia* sp. sponge sample collected near the Weizhou Island, Beibu Gulf of the South China Sea. In the chemical study of this strain, ten alkaloids (**1**–**10**), including four quinazoline-containing diketopiperazines (**2**–**5**), two butenolide derivatives (**11**, **12**), and four other metabolites (**13**–**16**) were obtained. Among them, **1**, **2**, and **11** were new compounds. Herein, we report the isolation, structural elucidation, and bioassay screening of all isolated compounds (Figure 1).

## 2. Results

The fungus *Penicillium* sp. SCSIO 41413 was fermented in rice medium, which was extracted with EtOAc to obtain crude extract after 30-day fermentation. Several chromatographic methods, including silica gel column and semi-preparative HPLC with octadecylsilyl (ODS) column, were used for isolation of these 16 compounds.

Compound **1** was obtained as a yellowish oil. The molecular formula was established as C_15_H_14_N_2_O_2_ (*m*/*z* 255.1131 [M + H]^+^) in the HR-ESI-MS spectrum. The ^1^H NMR (Table 1) spectrum displayed eight aromatic proton signals (*δ*_H_ 7.19, H-7; 6.62, H-8; 7.63, H-9; 6.65, H-10; 6.72, H-13; 6.70, H-15; 7.14, H-16; 6.74, H-17). The ^13^C NMR spectrum showed one sp^3^ methyl (*δ*_C_ 32.0, C-3), one sp^3^ methylene (*δ*_C_ 72.0, C-4), twelve sp^2^ aromatic carbons (*δ*_C_ 114.0, C-6; 133.2, C-7; 114.2, C-8; 127.3, C-9; 116.8, C-10; 146.4, C-11; 142.2, C-12; 116.8, C-13; 157.6, C-14; 115.4, C-15; 129.6, C-16; 116.8, C-17), and a conjugated carbonyl carbon (*δ*_C_ 162.5, C-1). The presence of an *ortho* substituted aromatic ring was established by detailed analysis of the 1D NMR data. The HMBC correlations (Figure 2) from H-7 to C-8/C-9/C-11, and from H-3 to C-1/C-4, suggested the 4-quinazolone ring system in the molecule. The NMR data of **1** were closely related to those of glycozolone A, a racemic natural quinazoline alkaloid [9]. The main difference was the appearance of 5-NH and 14-OH groups in **1** instead of 1-NCH_3_ and a hydrogen atom in glycozolone A, respectively, which was confirmed in ^1^H-^1^H COSY correlation of NH-5/H-4 and HMBC correlations from H-16 to C-12/C-14/C-17. Given the low measured specific optical rotation (${\left[\alpha \right]}_{D}^{25}-$3.9 (*c* 0.03, MeOH)) and a marginal CD effect (Appendix A) of **1**, it was suggested to be a racemic mixture, which was not subjected to chromatographic resolution due to limited sample quantities. Therefore, the structure of **1** was determined as 2-(3-hydroxyphenyl)-3-methyl-2, 3-dihydroquinazolin-4(1*H*)-one and was given trivial name polonimide E (**1**).

Compound **2** was obtained as a white amorphous powder. The molecular formula of C_21_H_25_N_3_O_6_ (11 degrees of unsaturation) was established by the positive high-resolution electrospray ionization mass spectroscopy (HR-ESI-MS) peak at *m*/*z* 416.1820 [M + H]^+^ (calcd 416.1816 for C_21_H_26_N_3_O_6_). The ^1^H and ^13^C NMR data collected in DMSO-*d*_6_ (Table 1) showed two methyl signals (*δ*_H/C_ 1.08/22.0, CH_3_-20; 1.05/22.3, CH_3_-21), four methylenes (*δ*_H/C_ 3.90, 3.78/65.7, CH_2_-22; 3.29/62.2, CH_2_-24; 2.37/29.5, CH_2_-16; and 2.13, 2.04/27.4, CH_2_-15), and eight methines (*δ*_H/C_ 7.84/134.8, CH-8; 7.69/126.8, CH-9; 8.13/125.3, CH-10; 7.53/126.3, CH-7; 6.22/127.2, CH-18; 3.57/69.1, CH-23; 5.19/54.3, CH-14; and 2.98/25.0, CH-19). Moreover, the ^13^C NMR spectrum also displayed three carbonyls (*δ*_C_ 165.3, C-1; 159.9, C-12; and 171.4, C-17) and four quaternary carbons (*δ*_C_ 126.7, C-3; 145.5, C-4; 147.0, C-6; and 119.7, C-11). With careful analyses of the 1D-NMR data, it was found that **2** shared the same quinazoline core as that of co-isolated polonimide A (**4**), a diketopiperazine alkaloid isolated from the fungus *Penicillium polonicum* [7]. Interestingly, the methoxy group in **4** was replaced by a glycerol moiety in **2**, which was confirmed by the ^1^H-^1^H COSY correlations of H_2_-22/H-23/H_2_-24 (Figure 2). The HMBC correlations (Figure 2) from H_2_-22 to C-17, C-23, and C-24; from H-23 to C-22 and C-24; and from H-24 to C-23 and C-22 also verified the above-mentioned glycerol moiety attached at C-17. Thus, the planar structure of **2** was determined and was named polonimide D (**2**).

The relative configuration (RC) of the Δ^3,18^ double bond in **2** was mainly determined by comparing the chemical shifts of CH-18, CH-19, and C-3 with those reported siblings, polonimides A (**4**) and B (**3**), aurantiomide C (**5**), RCs of which were determined by selective 1D nuclear overhauser effect (NOE) experiments [7]. The chemical shifts of CH-18 (*δ*_H/C_ 6.22/127.1), CH-19 (*δ*_H/C_ 3.00/25.0), and C-3 (*δ*_C_ 126.6) of **2** measured in DMSO-*d*_6_ were completely identical with those [CH-18 (*δ*_H/C_ 6.22/127.2), CH-19 (*δ*_H/C_ 2.98/25.0), and C-3 (*δ*_C_ 126.7)] of polonimide A (**4**) also measured in DMSO-*d*_6_ [7], suggesting *Z*-configuration of the Δ^3,18^ double bond in **2**. Moreover, the experimental ECD spectra (Figure 3) of **2** also showed good agreement with those reported both experimental and calculated ECD spectra of **3**–**5**, revealing the shared 14*S* absolute configuration (AC) among them, AC of which was reliably assigned by Marfey’s analysis [7]. The Mosher’s reaction approach for determining the absolute configuration of the chiral center C-23 turned to failure, thus it was unsolved owing to limited obtained quantities.

Compound **11** possessed the elemental composition of C_17_H_14_O_6_ (11 degrees of unsaturation) as established by the HRESIMS data (*m*/*z* 315.0867 [M + H]^+^). The ^1^H and ^13^C NMR data measured in DMSO-*d*_6_ (Table 2) showed six methines (*δ*_H/C_ 6.77/120.1, CH-18; 6.44/119.2, CH-12; 6.95/116.1, CH-14; 6.55/115.9, CH-8; 6.66/115.8, CH-11; and 6.80/115.6, CH-17), two methylenes (*δ*_H/C_ 4.69/70.8, CH_2_-5; 3.74/32.5, CH_2_-6), one carbonyl (*δ*_C_ 173.2, C-2), and eight quaternary carbons (*δ*_C_ 124.9, C-3; 160.2, C-4; 127.4, C-7; 145.5, C-9; 144.2, C-10; C-120.8, C-13; 145.2, C-15; and 145.9, C-16). Detailed analysis of the NMR data of **11** was closely related to those of **12** [10], which also suggested **11** was a butenolide derivative with two 1,2-disubstituted benzene moieties. The main difference was the hydroxyl substitution at C-9 and a methylene of CH_2_-5 in **11** instead of a hydrogen atom at C-9 and hydroxyl substituted methine in **12**, respectively, which was confirmed by HMBC correlations (Figure 2) from H-11, H-8 to C-9, H-5 (*δ*_H_ 4.69, s, 2H) to C-2, and C-3 in **11**. Hence, the structure of **11** was determined and was named as eutypoid F (**11**).

The thirteen known compounds were identified as polonimide B (**3**) [7], polonimide A (**4**) [7], aurantiomide C (**5**) [11], fructigenine A (**6**) [12], 3-*O*-methylviridicatin (**7**) [13], viridicatol (**8**) [14], arctosin (**9**) [15], cyclopenin (**10**) [16], 8-hydroxyhelvafuranone (**12**) [10], verrucosidinol acetate (**13**) [17], deoxyverrucosidin (**14**) [18], nordeoxyverrucosidin (**15**) [19], and aspterric acid methyl ester (**16**) [12], respectively, by comparison of their NMR and MS data with those reported in the literature. Notably, 8-hydroxyhelvafuranone (**12**) was also obtained as a racemic mixture due to the low measured specific optical rotation (${\left[\alpha \right]}_{D}^{25}+$3.1 (c 0.05, MeOH)) and a marginal CD effect (Appendix A).

All compounds were assessed for antibacterial activities against six pathogenic bacteria, including *Acinetobacter baumannii* (ATCC 19606), *Staphylococcus aureus* (ATCC 29213), *Enterococcus faecalis* (ATCC 29212), *Klebsiella pneumoniae* (ATCC 13883), *Escherichia coli* (ATCC 25922), and Methicillin-resistant *Staphylococcus aureus*, and eight phytopathogenic bacteria, including *Colletotrichum asianum* HNM408, *Colletotrichum gloeosporioides* HNM1003, *Colletotrichum acutatum HNMRC178, Fusarium oxysporum HNM1003, Pyricularia oryaza HNM 1003, Alternaria alternate, Curvularia australiensis,* and *Rhizoctonia solani,* using disc agar diffusion method. Two human prostate cancer cell lines, PC-3 (androgen receptor negative) and 22Rv1 (androgen receptor positive), were also used in the cytotoxicity test [20]. However, none of them showed obvious antibacterial or cytotoxic activities.

Because of the diverse biological activities reported with quinazoline-containing diketopiperazines and butenolide derivatives [7,8], further bioactivity screening is necessary, such as anti-tumor related enzymatic bioassay. Enzymes phosphatidylinositol 3-kinase (PI3K) and 6-phosphofructo-2-kinase/fructose-2,6-biphosphatase 3 (PFK-2/FBPase 3, PFKFB3), which played a significant role in the regulation of glycolysis in cancer cells as well as its proliferation and survival [21], were used for assessment of enzyme activities in our study. As a result, none of the compounds showed inhibition above 50% against PFKFB3 enzyme at 20 μM, while **11** and **12** displayed obvious inhibition against PI3K with IC_50_ values of 1.7 μM and 9.8 μM, respectively.

To obtain an insight into the molecular interactions between **11** and **12** and PI3K, molecular docking analysis was carried out. Butenolide derivatives **11** and **12** expressed the interaction with PI3K protein (PDB ID: 1E7U) perfectly, and the docking scores were −11.688 and −8.863, respectively. KWT (molecule of wortmannin) is a ligand in 1E7U designated by the RCSB. As shown in Figure 4A-C, phenolic hydroxy groups of **11** formed hydrogen bonds with the active site residues THR887, TYR867, and ALA885, and the ester group also interacted with VAL882 by hydrogen bond. Meanwhile, phenolic hydroxy groups of **12** formed three hydrogen bonds with residues THR887, TYR867, and ASP950. These results provide us rational explanation of the interactions between butenolide derivatives **11** and **12** and PI3K, which provides valuable information for further development of PI3K inhibitors.

Nuclear factor-κB (NF-κB) is a protein complex that controls transcriptional DNA, cytokine production, and cell survival, and is an important intracellular nuclear transcription factor [22]. The over-activation or defect of NF-κB can lead to the abnormal expression of various target cell genes, which is related to the inflammatory changes of many human diseases such as rheumatoid arthritis and heart and brain diseases. Therefore, inhibiting the NF-κB signal transduction pathway by drugs may become a means to treat many inflammatory diseases [23]. Seven compounds (**3**–**6**, **14**–**16**) were screened for their inhibitory activities of LPS-induced NF-κB activation in RAW264.7 at 10 µM, and four of them (**3**–**4**, and **14**–**15**) were revealed with the activity of varying strength (Figure 5). Compounds **4** and **15** showed significant inhibitory activity against LPS-induced NF-κB (*p* < 0.001).

## 3. Materials and Methods

### 3.1. General Experimental Procedures

The NMR spectra were obtained on a Bruker Avance spectrometer (Bruker, Billerica, MA, USA) operating at 500 and 700 MHz for ^1^H NMR, and 125 and 175 MHz for ^13^C NMR, using tetramethylsilane as an internal standard. High-resolution mass spectra were recorded on a Bruker miXis TOF-QII mass spectrometer (Bruker, Billerica, MA, USA). Optical rotations were measured on a PerkinElmer MPC 500 (Waltham, MA, USA) polarimeter. UV and ECD spectra were recorded on a Chirascan circular dichroism spectrometer (Applied Photophysics, Leatherhead Surrey, UK). The TLC and column chromatography (CC) were performed on plates precoated with silica gel GF254 (10–40 µm), and over silica gel (200–300 mesh) (Qingdao Marine Chemical Factory, Qingdao, China), Sephadex LH-20 (Amersham Pharmacia Biotech AB, Uppsala, Sweden), and semi-preparative HPLC using an ODS column (YMC-pack ODS-A, YMC Co., Ltd., Kyoto, Japan, 10 mm × 250 mm, 5 µm). All solvents employed were of analytical grade. The sea salt (Guangzhou Haili Aquarium Technology Company, Guangzhou, China) was a commercial product.

### 3.2. Fungal Strain

The fungal strain *Penicillium* sp. SCSIO 41413 was isolated from a sponge (*Callyspongia* sp.) sample which was collected near the Weizhou Island (Guangxi, China), Beibu Gulf of the South China Sea. This strain was stored on MB agar (malt extract 15 g, sea salt 10 g, H_2_O 1 L, PH 7.4–7.8) slants at 4 °C and deposited at Key Laboratory of Tropical Marine Bioresources and Ecology, Chinese Academy of Sciences. The ITS sequence region of the strain SCSIO 41413 was amplified by PCR, and rDNA sequencing showed that it shared significant homology to that of *Penicillium*. The rDNA sequence has 100% sequence identity to that of *Penicillium polonicum* (GenBank accession no. NR_103687.1), so it was designated as *Penicillium* sp. SCSIO 41413.

### 3.3. Fermentation and Extraction

Seed medium (malt extract 15 g, sea salt 10 g, distilled water 1000 mL) was inoculated with *Penicillium* sp. SCSIO 41413 and incubated at 25 °C for 72 h on a rotating shaker (180 rpm/s). The strain *Penicillium* sp. SCSIO 41413 was cultured in the flasks (×60) of rice medium (rice 200 g/flask, sea salt 7.0 g/flask, distilled H_2_O 200 mL/flask). These flasks were incubated statically at 25 °C under a normal day/night cycle. After 30 days, the rice medium was soaked in EtOAc (600 mL/flask), cut into small pieces, and sonicated for 20 min. Then, they were poured into fermentation vats, which were extracted with EtOAc four times and concentrated under reduced pressure to obtain a crude extract. The crude extract was suspended in MeOH and then partitioned with an equal volume of petroleum ether (PE) in order to remove the oil. At last, the MeOH solution was concentrated under reduced pressure to obtain a black crude extract (85.0 g).

### 3.4. Isolation and Purification

The crude extract was subjected to silica gel column chromatography (30 × 500 mm) using step gradient elution with CH_2_Cl_2_-MeOH (*v*/*v* 100:1, 80:1, 50:1, 20:1, 0:1, 500 mL each) to obtain ten subfractions (Frs.1–10) based on TLC analysis. Fr.4 was subjected to ODS CC (20 × 300 mm) with MeOH-H_2_O (*v*/*v*, 1:9, 3:7, 5:5, 7:3, 10:0, 200 mL each) to obtain ten subfractions (Fr.4.1-4.10). Then, Fr.4-5 and Fr.4-4 were purified by semi-preparative HPLC using an ODS column (YMC-pack ODS-A, YMC Co., Ltd., Kyoto, Japan, 10 × 250 mm, 5 µm) to obtain **5** (3.4 mg, 33% MeOH/H_2_O, 2.5 mL/min, *t*_R_ = 30.8 min), and **2** (3.5 mg, 37% MeCN/H_2_O, 2.5 mL/min, *t*_R_ = 12 min), **12** (11.1 mg, 43% MeOH/H_2_O, 2.5 mL/min, *t*_R_ = 21 min), **11** (7.1 mg, 35% MeCN/H_2_O, 2.5mL/min, *t*_R_ = 14.5 min), **1** (1.1 mg, 60% MeOH/H_2_O, 2.5 mL/min, *t*_R_ = 12 min), and **10** (19.5 mg, 38 % MeCN/H_2_O, 2.5 mL/min, *t*_R_ = 14.5 min). Fr.2 was subjected to ODS CC with MeOH-H_2_O (3:7-10:0, *v*/*v*) to obtain ten subfractions (Fr.2.1-2.10). Then, Fr.2-10 and Fr.2-6 were further purified by semi-preparative HPLC to obtain **14** (6.1 mg, 75% MeOH/H_2_O, 2.5 mL/min, *t*_R_ = 19 min), and **15** (7.6 mg, 75% MeOH/H_2_O, 2.5 mL/min, *t*_R_ = 18 min), **7** (2.8 mg, 45% MeCN/H_2_O, 2.5 mL / min, *t*_R_ = 27 min), **4** (19.2 mg, 45 % MeCN/H_2_O, 2.5 mL/min, *t*_R_ = 30 min), and **16** (10.7 mg, 64% MeOH/H_2_O, 2.5 mL/min, *t*_R_ = 30 min). Fr.5 and Fr.3 were purified by HPLC to obtain **3** (3.1 mg, 47 % MeCN/H_2_O, 2.5 mL/min, *t*_R_ = 12 min), **6** (3.9 mg, 53 % MeCN/H_2_O, 2.5 mL/min, *t*_R_ = 24 min), **13** (2.5 mg, 53% MeCN/H_2_O, 2.5 mL/min, *t*_R_ = 18 min), **8** (7.5 mg, 60% MeOH/H_2_O, 2.5 mL/min, *t*_R_ = 10 min), and **9** (5.6 mg, 38% MeCN/H_2_O, 2.5 mL/min, *t*_R_ = 14.5 min).

*Polonimide E* (**1**): yellow oil; ${\left[\alpha \right]}_{D}^{25}$ −(*c* 0.03, MeOH); UV (MeOH) λ_max_ (log*ε*): 204 (3.00), 222 (3.08) nm; ECD (0.3 mg/mL, MeOH) λ_max_ (Δε) 200 (+8.38), 202 (−0.89), 212 (+2.62), 219 (−2.56); IR (film) *ν*_max_ 3288, 2927, 1625, 1608, 1487, 1456, 1278, 1153, 1024 cm^−1^. ^1^H NMR (DMSO-*d*_6_, 700 MHz) and ^13^C NMR (DMSO-*d*_6_, 176 MHz) data, see Table 1; HRESIMS *m/z* 255.1131 [M + H]^+^ (calcd for C_15_H_15_N_2_O_2_ 255.1128), 277.0953 [M + Na]^+^ (calcd for C_15_H_14_N_2_O_2_Na 277.0947).

*Polonimide D* (**2**): yellow oil; ${\left[\alpha \right]}_{D}^{25}$ −11.1 (c 0.10, MeOH); UV(MeOH) λ_max_ (logε): 209 (2.85), 306 (1.77) nm, ECD (0.3 mg/mL, MeOH) λ_max_ (Δε) 202 (−9.93), 225 (+9.73), 249 (−6.50); IR (film) *ν*_max_ 2960, 2868, 1716, 1682, 1582, 1560, 1394, 1336, 1292, 1247, 1168, 1024 cm^−1^. ^1^H NMR (DMSO-*d*_6_, 500 MHz) and ^13^C NMR (DMSO-*d*_6_, 125 MHz) data, see Table 1; HRESIMS *m/z* 416.1820. [M + H]^+^ (calcd for C_21_H_26_N_3_O_6_ 416.1816), 438.1639 [M + Na]^+^ (calcd for C_21_H_25_N_3_O_6_Na, 438.1636).

*Eutypoid F* (**11**): white powder; UV (MeOH) λ_max_ (log*ε*): 204 (3.98), 290 (2.93) nm. IR (film) *ν*_max_ 2945, 2833, 2260, 2129, 1732, 1602, 1520, 1362, 1283, 1199, 1115, 1020 cm^−1^.^1^H NMR (DMSO-*d*_6_, 500 MHz) and ^13^C NMR (DMSO-*d*_6_, 125 MHz) data, see Table 2; HRESIMS *m/z* 315.0867 [M + H]^+^(calcd for C_17_H_15_O_6_ 315.0863), 337.0689 [M + Na]^+^ (calcd for C_17_H_14_O_6_Na 337.0863).

### 3.5. Bioassay

The antibacterial activity was assessed using the K-B disc agar diffusion method [24]. Compounds **1**–**16** were tested for antibacterial activities against six pathogenic bacteria, *Acinetobacter baumannii* (ATCC 19606), *Staphylococcus aureus* (ATCC 29213), *Enterococcus faecalis* (ATCC 29212), *Klebsiella pneumoniae* (ATCC 13883), *Escherichia coli* (ATCC 25922), and Methicillin-resistant *Staphylococcus aureus* (MRSA), in which ampicillin and gentamicin were used as a positive control for gram-positive and gram-negative bacteria, respectively, and eight phytopathogenic bacteria, *Colletotrichum asianum* HNM408, *Colletotrichum gloeosporioides* HNM1003, *Colletotrichum acutatum* HNMRC178, *Fusarium oxysporum* HNM1003, *Pyricularia oryaza* HNM 1003, *Alternaria alternate*, *Curvularia australiensis*, and *Rhizoctonia solani*, in which nystatin and methanol were used as the positive control and negative control, respectively.

Two human prostate cancer cell lines, PC-3 (androgen receptor negative) and 22Rv1 (androgen receptor positive), were used in the cytotoxicity tests, and cell viability was analyzed by MTT assay as previously described [20].

Enzyme activities of those compounds against PI3K [25] and PFKFB3 [21] were evaluated using the methods reported. For preliminary screening, the final concentration of 20 μM was used. Further PI3K enzyme activity assay was performed to determine the 50% inhibition concentration (IC_50_) values of **11** and **12**, with the concentrations of 80, 40, 20, 10, 5, 2.5, 1.25, and 0.625 μM.

The inhibitory activities of LPS-induced NF-κB activation in RAW264.7 cells were evaluated as detected by luciferase reporter gene assay as described previously [20]. In other words, the RAW264.7 cells stably transfected with luciferase reporter gene were placed in 96-well plates and pretreated with tested compounds (10 µM) and BA Y11-7082 (NF-κB inhibitor as positive control, 5 µM, Sigma-Aldrich) for 30 min. Then, they were stimulated with 5 µg/mL LPS for 8 h. The cells were collected, and luciferase activities of the triplicate tests were measured using the luciferase assay system (Promega, Madison, WI, USA).

### 3.6. Molecular Docking

The Schrödinger 2017-1 suite (Schrödinger Inc., New York, NY, USA) was employed to perform the docking analysis. The crystal structure of PI3K (PDB code: 1E7U) [26] obtained from the Protein Data Bank was used as a starting model with all of the waters and the *N*-linked glycosylated saccharides removed and was constructed following the Protein Prepare Wizard workflow in Maestro 11-1. The prepared ligands were then flexibly docked into the receptor using the induced-fit module with the default parameters. The figures were generated using PyMol molecular graphics software (Schrödinger 2017-1, Schrödinger Inc., New York, NY, USA).

## 4. Conclusions

In summary, chemical investigation of the marine fungus *Penicillium* sp. SCSIO 41413, which was isolated from a Beibu Gulf sponge sample, led to the isolation of two new alkaloids, polonimides E (**1**) and D (**2**), a new butenolide derivative, eutypoid F (**11**), and thirteen known compounds (**3**−**10**, **12**−**16**). Their planar structures and absolute configurations were elucidated by detailed NMR, MS spectroscopic analyses, and measured and calculated ECD analyses. Butenolide derivatives **11** and **12** exhibited obvious inhibitory against the enzyme PI3K with IC_50_ values of 1.7 μM and 9.8 μM, respectively, while **4** and **15** exhibited obvious inhibitory activity against LPS-induced NF-κB activation in RAW264.7 cells at 10 µM. The molecular docking with PI3K protein was also performed to understand the inhibitory activity. This study provides valuable information for further development of PI3K or NF-κB inhibitors.

## Figures and Tables

**Figure 1 marinedrugs-21-00027-f001:**
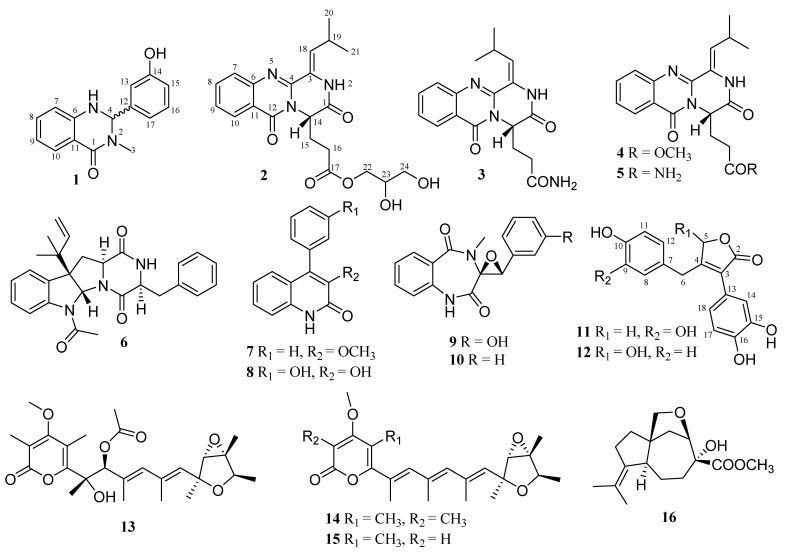
Chemical structures of **1**–**16**.

**Figure 2 marinedrugs-21-00027-f002:**
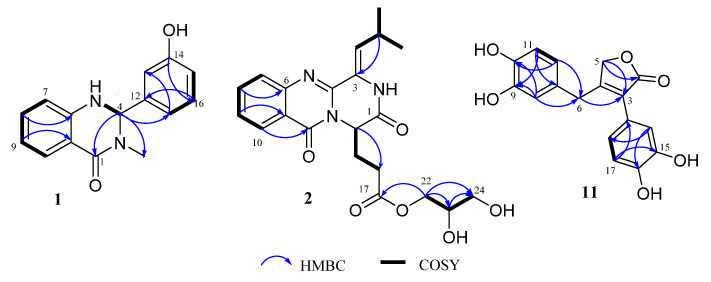
Key HMBC and ^1^H-^1^H COSY correlations of **1**, **2**, and **11**.

**Figure 3 marinedrugs-21-00027-f003:**
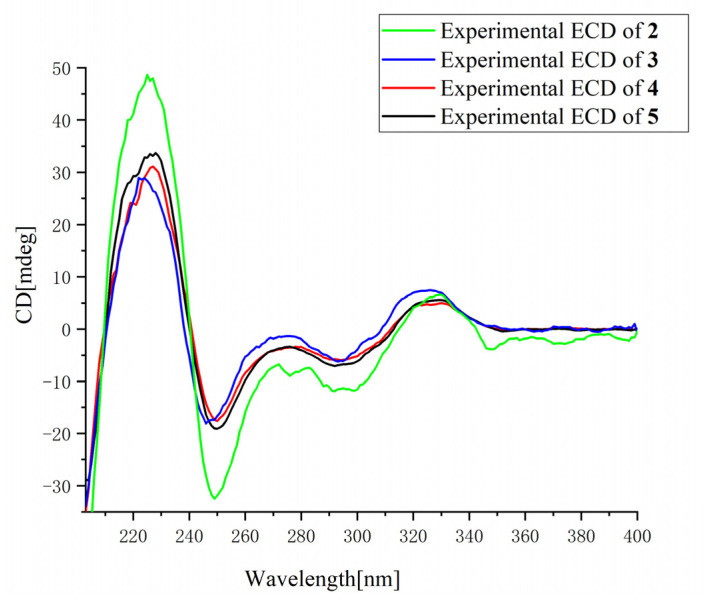
Experimental ECD spectra of **2**–**5**.

**Figure 4 marinedrugs-21-00027-f004:**
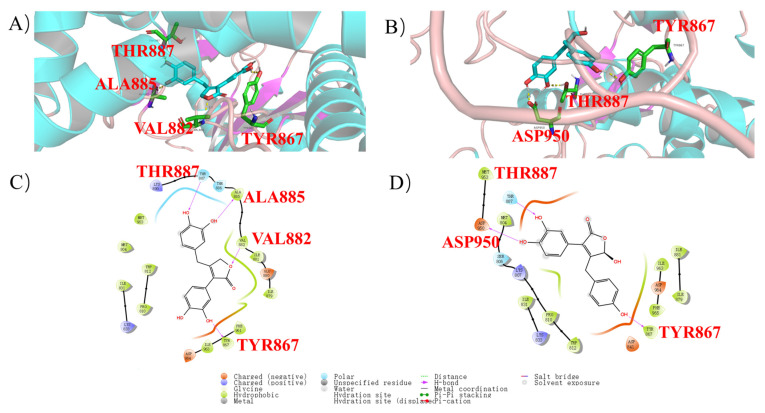
Molecular docking of **11** and **12** with PI3K (PDB ID: 1E7U). (**A**) Binding sites of the molecular **11** with the 1E7U crystal structures (part). (**B**) Binding sites of the molecular **12** with 1E7U crystal structures (part). (**C**) The interaction details of the predicted binding mode of **11** with the 1E7U. (**D**) The interaction details of the predicted binding mode of **12** with the 1E7U.

**Figure 5 marinedrugs-21-00027-f005:**
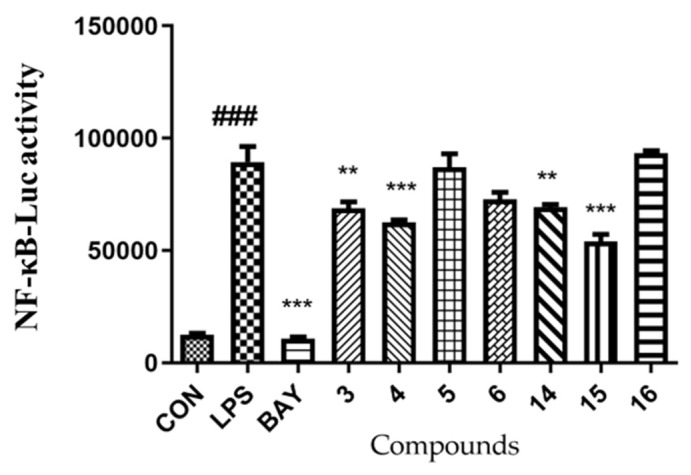
The inhibitory activity of compounds **3**–**6**, **14**–**16** against LPS-induced NF-κB activation in RAW264.7 cells at 10 µM. *n* = 3. ^###^ *p* < 0.001 vs. control group (untreated); *** *p* < 0.001, ** *p* < 0.01 vs. LPS-induced group. BAY (BAY11-7082 treated, positive control).

**Table 1 marinedrugs-21-00027-t001:** The ^1^H (500 MHz) and ^13^C NMR (125 MHz) data of **1** and **2** (TMS, *δ* in ppm, DMSO-*d*_6_).

Position	1		2	
	*δ*_C_ Type	*δ*_H_ Mult. (*J* in Hz)	*δ*_C_ Type	*δ*_H_ Mult. (*J* in Hz)
1	162.5, C		165.3, C	
3	32.0, CH_3_	2.84, s	126.7, C	
4	72.0, CH	5.72, d, (2.4)	145.5, C	
5		7.26, d, (2.4)		
6	114.0, C		147.0, C	
7	133.2, CH	7.19, dd, (7.7, 1.6)	126.3, CH	7.53, m
8	114.2, CH	6.62, m	134.8, CH	7.84, ddd, (8.2, 7.1, 1.5)
9	127.3, CH	7.63, dd, (7.7, 1.6)	126.8, CH	7.69, d, (8.2)
10	116.8, CH	6.65, dd, (7.7, 1.6)	125.3, CH	8.13, dd, (8.2, 1.5)
11	146.4, C		119.7, C	
12	142.2, C		159.9, C	
13	112.8, CH	6.72, t, (2.1)		
14	157.6, C		54.3, CH	5.19, m
15	115.4, CH	6.70, ddd, (7.8, 2.1, 1.3)	27.4, CH_2_	2.13, dq, (14.0, 7.0)2.04, dq, (14.0, 7.0)
16	129.6, CH	7.14, t, (7.8)	29.5, CH_2_	2.37, q, (7.0)
17	116.8, CH	6.74, dt, (7.8, 1.3)	171.4, C	
18			127.2, CH	6.22, d, (10.3)
19			25.0, CH	2.98, m
20			22.0, CH_3_	1.08, d, (6.5)
21			22.3, CH_3_	1.05, d, (6.5)
22			65.7, CH_2_	3.90, ddd, (10.4, 6.6, 4.0)3.78, ddd, (10.4, 6.6, 4.0)
23			69.1, CH	3.57, q, (5.7)
24			62.2, CH_2_	3.29, dd, (5.7, 4.0)

**Table 2 marinedrugs-21-00027-t002:** The ^1^H (500 MHz) and ^13^C (125 MHz) NMR data of **11** (TMS, *δ* in ppm, DMSO-*d*_6_).

Position	11	
	*δ*_C_ Type	*δ*_H_ Mult. (*J* in Hz)
2	173.2, C	
3	124.9, C	
4	160.2, C	
5	70.8, CH_2_	4.69, s
6	32.5, CH_2_	3.74, s
7	127.4, C	
8	115.9, CH	6.55, d, (2.2)
9	145.5, C	
10	144.2, C	
11	115.8, CH	6.66, d, (8.0)
12	119.2, CH	6.44, dd, (8.0, 2.2)
13	120.8, C	
14	116.1, CH	6.95, d, (2.0)
15	145.2, C	
16	145.9, C	
17	115.6, CH	6.80, d, (8.1)
18	120.1, CH	6.77, dd, (8.1, 2.2)

## Data Availability

Not applicable.

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
