# Peer review of "Two New Alkaloids and a New Butenolide Derivative from the Beibu Gulf Sponge-Derived Fungus Penicillium sp. SCSIO 41413"

_marinedrugs, 2022, doi:10.3390/md21010027_

Round 1

Reviewer 1 Report

This submitted manuscript reported a detailed chemical study on the Beibu Gulf sponge-derived fungus Penicillium sp. SCSIO 41413, leading to the discovery of two new alkaloids, polonimides D (1) and E (2), and a new butenolide derivative, eutypoid F (3), together with thirteen known compounds 416. Moreover, the absolute configuration of the new compound 1 was determined by ECD calculation. Importantly, some of the isolates exhibited significant inhibitory against PI3K and NF-κB. Additionally, molecular docking was performed to give an insight into the molecular interactions between the active compounds and PI3K. This work is solid, and these new findings are intriguing. I will recommend it after minor revision.

Comments:

1. As known to all, it is challenging to determine the absolute configuration of a flexible molecule. There was a glycerol moiety in compound 1, but there was no indicator of the absolute configuration of the chrial center C-23. Had you tried any approach to determine it?

2. Had the absolute configurations of the known compounds 57 been determined before? The ECD calculation work was shown in Figure 3, but there was lack of specification.

3. There were twelve sp2 aromatic carbons in the chemical structure of compound 2, not ‘eight sp2 aromatic carbons’ (P4L83). In fact, there were eight sp2 aromatic methines. And as the 1H–1H COSY correlation of 1-NH/H-2 was a key support for the structure elucidation, please add the 1H NMR chemical shift of 1-NH in Table 2.

4. What’s the specific optical rotation datum of compound 2? Was there any possibility to determine the absolute configuration of the chrial center C-2?

5. What’s the ligand used in the docking experiment? Please add the summary of the docking results in the Abstract and Conclusion.

6. Please list the names of the six pathogenic bacteria and eight phytopathogenic bacteria on P5, although they were shown in the Experimental section.

7. It is better to add the name of the sponge, from which the fungal strain Penicillium sp. SCSIO 41413 was isolated.

8. Please add the ECD calculation and molecular docking protocol in the Experimental section.

Others:

1. P1L21 & P1L42: typo errors ‘4~16    416  1~3    13

2. P1L25: ‘obvious effect activity    obvious inhibitory activity

3. Please keep the 1H NMR data ‘7.53 (P1L59) and ‘6.62 (P4L81) consistent with those ‘7.56–7.49’ and ‘6.63–6.61’ recorded in Tables 1 and 2, respectively.

4. P4L99: ‘One ester    one carbonyl

5. References: Please use the abbreviations of all journal names, and revise the pages of Ref. 13 as ‘1329-1332’.

Reviewer 2 Report

This manuscript reports two new alkaloids and a new butanolide from a fungal strain derived from a sponge collected in Beibu Gulf. When the three new compounds were discovered, 13 known compounds were also identified from the fungal strain. The planar structures of the new compounds were elucidated by spectroscopic analysis and their biological activities were evaluated along with the known compounds. In respect to reporting natural products from a Beibu Gulf sponge-derived fungus, this work is valuable as a chemical reference for natural products from Beibu Gulf. In addition, The buetenolide derivatives including the new compound, eutypoid F, displayed significant inhibitory activity against the kinase PI3K, verifying the importance of the work. However, the reviewer thinks that this manuscript should be improved for further consideration as a contribution to Marine Drugs. The reviewer suggests that the authors should consider the following point for revision.

(1) Because this work started from a sponge sample collected from Beibu Gulf, it would be great to describe the sponge distribution in Beibu Gulf, if there are suitable previous references. In addition, the sponge providing the fungal strain should be phylogenetically analyzed in the experimental section.

(2) The absolute configuration of C-2 in compound 2 should be determined. In compound 3, C-5 stereogenic center should be established configurationally. No experiment has been done for these two chiral centers.

(3) It would be better to divide Table 1 into two tables for compounds 1 and 3, respectively, because they belong to different structural classes.

(4) Please explain the meaning of the docking scores.

(5) In the isolation and purification experimental section, the column dimensions of column chromatography and HPLC should be provided. Solvent elution volumes should be also notified.

(6) In the physicochemical data of the new compounds, IR data are missing. Optical rotation for compound 2 should be provided. UV data should be listed from a short wavelength.

(7) Use the correct reference format. For example, “.” should be used after an abbreviated word. Besides the point, there are many errors.

Reviewer 3 Report

 This manuscript reported three new and 13 known compounds from a marine sponge-derived fungi have been proven to be a prolific source of bioactive natural 18 products. Two butenolide derivatives 3 and 4 exhibited inhibito effect against the enzyme PI3K with IC50 values of 1.7 μM and 9.8 μM, respectively. The authors need to improve the manuscript before publishing.

1.Fig. 1: Please present the structures in a better sequence. For instance, list all the steroids to together.

2. The author claimed the absolute configurations were solved. However, the absolute configurations of C-23 in 1, C-6 in 2 have not been mentioned.

3.The IR spectra of new compounds were not provided.

Reviewer 4 Report

In this manuscript, the authors reported two new alkaloids and one butenolide from a sponge-derived fungus Penicillium sp. The authors wrote the manuscript so simple. The stereochemistry of the new compounds normally need to be analyzed by 2D NMR and some chemical derivatization. However, I feel that the method described by the authors is not sufficient to explain the stereochemistry of compounds 1 and 3. The comments for this manuscript are as follows.

1. The authors need to supplement some introduction on the two types of the new compounds in the "Introduction" section, such as their characteristics of chemistry and their bioactivities.

2. Every carbon position of compounds 1-3 needs to be marked in figure 1, so that the readers can distinguish each carbon very clearly through the figure.

3. The structure of compound 1 is similar to that of 6. The authors explained the configuration of C-14 by the ECD method, but I cannot tell which the calculated curves have the same cotton effect with that of the experimental result. 14S-1 and 14R-1 showed different cotton curves at 220-230 nm and 320-330 nm from those of compound 1. Please make sure the correct configuration of C-14 in 1.

4. Further, the configuration of C-23 hasnot been elucidated in this manuscript. I suggest the author to do the Mosher's reaction to resolve the secondary hydroxyl group at C-23.

5. In the structural eucidation of compound 1, the authors also didn't explain the configuration of the double bond at C-3/C-18. Why did you determine it as "Z" configuration without any explanation, even though the known compounds of 5 and 6 have the different configuration at C-18?

6. Compound 3 has a similar structure to 4,, but the authors didn't elucidate their structural characteristics of both compounds, and didn't compare the differences between the two compounds. I suggest to compare their differences of spectral data of the structures so that they can explain the structure clearly. 

7. The NMR data of the proton at C-7 is different from that showed in Table 1. Please check.

8. The structure of compound 3 includes two benzene rings. Do the author need to determine a NOESY spectrum to explain their configuration in the structure?

9. In the experimental section, the authors should write the procedure for each bioactivity assay adequately. The protocal for the assays are so simple.

10. All the compounds in this manuscript didn't show obvious cytotoxicity. However, the authors evaluated the inhibitory activity against PI3K of compounds 3 and 4. Since PI3k is related to cancer, why did you consider to evaluate their activity against PI3K? Please explain it in the manuscript.

Round 2

Reviewer 2 Report

The revised version of the manuscript clarified the points that the reviewer raised. The reviewer suggests that this manuscript should be accepted in Marine Drugs.
